# Hydrothermal Aging of an Epoxy Resin Filled with Carbon Nanofillers

**DOI:** 10.3390/polym12051153

**Published:** 2020-05-18

**Authors:** Tatjana Glaskova-Kuzmina, Andrey Aniskevich, George Papanicolaou, Diana Portan, Aldobenedetto Zotti, Anna Borriello, Mauro Zarrelli

**Affiliations:** 1Institute for Mechanics of Materials, University of Latvia, LV-1004 Riga, Latvia; andrey.aniskevich@pmi.lu.lv; 2Composite Materials Group, Department of Mechanical Engineering & Aeronautics, University of Patras, GR-26500 Patras, Greece; gpapan@upatras.gr (G.P.); portan@upatras.gr (D.P.); 3Institute for Polymers, Composites and Biomaterials, National Research Council of Italy, 80055 Portici, Italy; aldobenedetto.zotti@unina.it (A.Z.); anna.borriello@cnr.it (A.B.); mauro.zarrelli@cnr.it (M.Z.)

**Keywords:** epoxy resin, nanocomposite, carbon nanotubes, carbon nanofibers, environmental degradation, flexural properties, modeling, property prediction model

## Abstract

The effects of temperature and moisture on flexural and thermomechanical properties of neat and filled epoxy with both multiwall carbon nanotubes (CNT), carbon nanofibers (CNF), and their hybrid components were investigated. Two regimes of environmental aging were applied: Water absorption at 70 °C until equilibrium moisture content and thermal heating at 70 °C for the same time period. Three-point bending and dynamic mechanical tests were carried out for all samples before and after conditioning. The property prediction model (PPM) was successfully applied for the prediction of the modulus of elasticity in bending of manufactured specimens subjected to both water absorption and thermal aging. It was experimentally confirmed that, due to addition of carbon nanofillers to the epoxy resin, the sorption, flexural, and thermomechanical characteristics were slightly improved compared to the neat system. Considering experimental and theoretical results, most of the epoxy composites filled with hybrid carbon nanofiller revealed the lowest effect of temperature and moisture on material properties, along with the lowest sorption characteristics.

## 1. Introduction

Currently, an increasing number of the most diverse engineering structures, including critical parts, are made of polymers and polymer composites in many industrial sectors. The performance and reliability of the structural components operated in different climatic conditions are highly determined by the physical and mechanical properties of the constituents. At the same time, the properties of polymers and polymer composites are different from those properties of the conventionally applied materials, such as metals, alloys, ceramics, etc. This makes it difficult to predict the performance and durability of the materials. In fact, first, there may be a strong dependence of material characteristics on the operating conditions (temperature, loading, time, and other factors) and, secondly, significant variability of the physical properties may exist due to environmental aging of polymers and polymer composites [1,2]. Generally, the environmental aging of polymers and polymer composites may be considered as an accumulation process of different damages, such as surface microcracks, bulk defects and pores, delamination, etc., arising from the action of atmospheric moisture, UV radiation, temperature, and other factors [3,4]. Obviously, an increase in the defectiveness of the materials leads to a degradation of their physical/mechanical characteristics.

Epoxy resins, which are widely applied for different structural applications, may have both reversible and irreversible changes of their physical and chemical properties due to environmental effects (e.g., moisture, elevated temperature, and UV radiation) [5,6]. Reversible changes result in plasticization of the epoxy resins and they occur due to hydro/hygrothermal exposure while irreversible changes result in chemical reactions (e.g., hydrolysis), leaching of partially uncured molecules, and post-curing [5,7].

As a matter of fact, the overall properties of the composite material depend on numerous parameters. Among them, the main are the properties of the polymer matrix and the filler, ab initio, shape and size of the filler particles, as well as the state of dispersion and the amount of the agglomeration [8,9]. Moreover, the adhesion bond between polymer matrix and filler particles, which is related to the interaction energy at the interphase, is also important and highly affects the physical/mechanical properties of the composites. More precisely, due to restriction of the segmental mobility of polymer macromolecules in the area between polymer matrix and filler, an extra phase, also called boundary interphase, appears. For the boundary interphase the structural microdefects, such as voids, microcracks, and impurities, are typical and may lead to localization of stress concentrations and, subsequently, to a preliminary breakage of the material.

In contrast to traditional microcomposites, the polymer nanocomposites (PNC) consist of a polymer matrix filled with nanofillers, characterized by a much higher polymer-filler contact area (10–100x). It mainly serves as an interface to transfer stress but also facilitates agglomeration of the nanoparticles. Moreover, due to large aspect ratio of the carbon nanoparticles, the required segregation to get good dispersion of nanoparticles in the matrix is the main issue and also a great challenge still to be solved exhaustively [8]. It is important to highlight that the formation of particle agglomerates, filler clusters, and/or microvoids may result in a nonhomogeneous stress distribution and subsequent localization of stress concentrations [10]. The addition of nanofillers such as carbon nanotubes (CNT), carbon nanofibres (CNF), and graphene in the case of their homogeneous dispersion could improve the environmental stability of mechanical, thermal, and electrical properties [3,6,11,12,13]. Moreover, in comparison with the conventional microcomposites, it may result in lightweight products and supply additional functionality to nonconductive polymer resins [14]. Though, up to date the efficiency and safety of their application in outdoor conditions are unclear due to their possible environmental aging caused by moisture and temperature effects [1]. Therefore, the most significant environmental effects on the operational properties of the polymer nanocomposites should be considered for the possible broadening of their outdoor applications.

Previously, it was reported that the extent of the degradation of elastic and viscoelastic properties of the epoxy resin filled with different amounts of CNT due to moisture [12] and temperature [13] effects was lower in comparison with the respective properties’ degradation for the neat epoxy resin. This was attributed to the restriction of the mobility for polymer chains due to high aspect ratio of stiff CNT. Though, in the case of agglomeration of nanofiller particles within polymers, the improvement of mechanical properties of epoxy matrix subjected to elevated temperatures and humidities may be diminished.

The aim of this work was to establish the effect of environmental exposure on the durability of neat epoxy resin and filled with carbon nanofillers, CNT, CNF, and hybrid nanofiller (HN), which was evaluated by performing mechanical and thermomechanical characterization during and after water absorption and/or thermal heating. The environmental effects (moisture and temperature) on the modulus of elasticity in bending (hereinafter, elastic modulus) were evaluated by using the property prediction model (PPM), which was previously successfully applied for the prediction of elastic modulus and strength of epoxy resin filled with micro- and nanoparticles of TiO_2_ as a function of filler concentration [15]. In the current paper and for the first time, PPM was developed and applied as a function of duration of environmental aging.

## 2. Materials and Methods

### 2.1. Materials

A monocomponent epoxy resin, RTM6 (Hexcel Composites, Stamford, Connecticut, USA), was used as a composite matrix. The system was filled with two different carbon nanofillers, multiwall CNT, Nanocyl 7000 (Nanocyl, Belgium), and CNF, SA 719781 (Sigma-Aldrich, St. Louis, MO, USA). Data of materials’ characteristics taken from manufacturers’ datasheets [16,17] are summarized in Table 1. According to Table 1, the aspect ratio of both carbon nanofillers was almost the same (150) while the average diameter and length of CNF were almost 13 times higher in comparison with those for CNT. Additionally, a hybrid nanofiller comprised of CNT and CNF (1:1 by weight) was used. The samples of neat epoxy and PNC at certain electrical conductivity of approx. 0.01 S/m required to provide effective structural health monitoring were prepared. Thus, the epoxy resin was filled with different amounts of single and hybrid carbon nanofillers, respectively, 0.05 wt. % of CNT, 0.3 wt. % of CNF, and 0.1 wt. % of HN for the characterization of the flexural and thermomechanical properties before, during (in 1 week), and after (in 4 weeks) the environmental aging. For each material type and test carried out in the current study the minimal amount of five specimens was assured. Thus, the values presented on the graphs of the paper correspond to the average values and their standard deviations presented as error bars.

### 2.2. Preparation of the Test Samples

High-shear disperser Ultra Turrax T25 and a heating plate, RCT basic (IKA, Staufen im Breisgau, Germany), were used for the dispersion of carbon nanofillers in the epoxy matrix. For all PNC, the dispersion process was the same as follows: Mixing for 30 min at a speed 20 krpm at 90 °C. This mixing procedure was based on the previously optimized solution [18]. Mixing was followed by degassing in vacuum oven (Memmert VO400, Schwabach, Germany) at 90 °C until total removal of air bubbles, curing at 160 °C for 90 min, and then post-curing at 180 °C for 2 h, as recommended by the RTM6 supplier. All materials were poured in and cured in steel moulds of round shape with diameter of 100 mm and thickness of 3 mm. At the end of the manufacturing phase, samples where achieved by cutting the circular plate into bar-shaped specimens. The average sizes of the specimens computed by taking measures in three different location along the axis were: 80 ± 2 mm, 10 ± 1 mm, and 3.0 ± 0.2 mm, respectively, for length, width, and thickness. All specimens were polished before the testing in order to minimize the effect of surface roughness on the water sorption kinetics.

### 2.3. Hydrothermal Aging

The hydrothermal aging was performed via water absorption at 70 °C and heating at the same temperature to investigate how such parallel procedures affect material properties of the epoxy resin and the PNC. The tested specimens were fully immersed in distilled water and periodically weighed by using XS205DU balance (Mettler Toledo, Columbus, OH, USA) with a precision of ± 0.05 mg to obtain the kinetics of moisture sorption. Water sorption at 70 °C lasted 4 weeks, when all samples reached the equilibrium moisture content with almost no change in mass (Δ*w* = ± 0.01%). Following immersion, specimens were removed at pre-determined times in order to weigh the percentage water “uptake” in accordance with the standard of American Society for Testing and Materials (ASTM) D5229. The percentage mass change *w_t_* (%) was determined by using formula:(1)$$w_{t}(\%)=\frac{m(t)-m_{0}}{m_{0}}\times 100$$
where *m*(*t*) is the measured mass at time *t* and *m*_0_ the initial dry mass, respectively.

### 2.4. Experimental Methods

The quantitative analysis of filler dispersion was performed by using an optical microscope, Olympus BX51 (Japan), as previously reported in [19]. By using *ImageJ* software it was possible to get binary images of optical micrographs and by applying iso-data algorithm to evaluate the area of separated filler particles. Finally, the distribution by particle size (area) was assessed for all studied materials by using histogram analysis.

Three-point bending tests were performed to characterize the mechanical properties for the neat epoxy and filled with carbon nanofillers. According to ASTM D790, the support span of 56 mm and a strain rate of 1.5 mm/min were chosen and applied for the test samples by using Zwick 2.5 machine (Zwick Roell Group, Ulm, Germany). From the stress-strain curves, the main flexural characteristics were evaluated: Elastic modulus, flexural strength, and maximal deformation. These results were also discussed in [19]. Dynamic mechanical analysis (DMA) of the samples before and after hydrothermal aging was performed by using Mettler Toledo DMA/SDTA861 (USA) for the evaluation of the effect of hydrothermal aging on the thermomechanical properties of the epoxy and the PNC. The testing procedure was a temperature scan from 30 to 280 °C at heating rate 3 K/min with an applied tensile force of 4 N at a frequency of 10 Hz.

### 2.5. Modeling of Property Degradation

Property prediction model has been developed [15] for the estimation of the composite property-value variation with filler concentration (*C_f_*) in particulate composites as well as with any parameter affecting its behavior, such as water absorption concentration, time of immersion, etc. For the model application, only two experimental points are required. In the case where the parameter affecting composite behavior is filler-volume fraction, the procedure reported below was applied.

The first experimental point (*C*_1_, *P*_1_) should represent the behavior of the composite having very low filler-volume fraction, but the second one (*C*_2_, *P*_2_) should represent the behavior of the composite having high filler-volume fraction. It should be noted that at very low filler loading the behavior of the composite was mostly affected by the adhesion between filler and matrix, but at high filler content the dispersion level played the major role.

The main model assumptions were: (1) The composite property value, *P_c_*, was mainly affected by filler-matrix adhesion and filler particle dispersion achieved by the manufacturing process system and (2) the composite property variation with filler loading, *C_f_*, could be characterized by using a second order polynomial function:(2)$$P_{c}=AC_{f}^{2}+BC_{f}+P_{m}$$
where *P_m_* is the property of the matrix.

Having two experimental points, (*C*_1_, *P*_1_) and (*C*_2_, *P*_2_), then
(3)$$A=\frac{P_{2}-P_{m}}{C_{2}\left(C_{2}-C_{1}\right)}-\frac{P_{1}-P_{m}}{C_{1}\left(C_{2}-C_{1}\right)}$$
(4)$$B=\frac{\left(P_{1}-P_{m}\right)C_{2}}{C_{1}\left(C_{2}-C_{1}\right)}-\frac{\left(P_{2}-P_{m}\right)C_{1}}{C_{2}\left(C_{2}-C_{1}\right)}$$

Setting now,
(5)$$\lambda =\frac{\left(P_{2}-P_{m}\right)}{P_{f}C_{2}\left(C_{2}-C_{1}\right)}$$
and
(6)$$\kappa =\frac{\left(P_{1}-P_{m}\right)}{P_{f}C_{1}\left(C_{2}-C_{1}\right)}$$

*A* and *B* can be written as:(7)$$A=\left(\lambda -\kappa \right)P_{f}$$
and
(8)$$B=\left(\kappa C_{2}-\lambda C_{1}\right)P_{f}$$
where *P_f_* is the property of the filler.

Then, relationship (2) can be written as: (9)$$P_{c}=\left(\lambda -\kappa \right)P_{f}C_{f}^{2}+\left(\kappa C_{2}-\lambda C_{1}\right)P_{f}C_{f}+P_{m}$$

The *κ*-parameter represents the adhesion coefficient because it was influenced by the low filler concentration (*C*_1_) and the property value (*P*_1_), i.e., point (*C*_1_, *P*_1_), when the behavior of the composite was mostly affected by the adhesion between filler and matrix. Analogously, the *λ*-parameter represents the dispersion coefficient as it relates to high filler concentration (*C*_2_) and the corresponding property value (*P*_2_), i.e., point (*C*_2_, *P*_2_).

When the elastic modulus of the composite *E_c_* = *f*(*C_f_*), Equation (2) is written as follows:(10)$$E_{c}=\left(\lambda -\kappa \right)E_{f}C_{f}^{2}+\left(\kappa C_{2}-\lambda C_{1}\right)E_{f}C_{f}+E_{m}$$

As mentioned above, it was assumed in the model that the variation of the modulus of particulates with filler concentration was mainly affected by the coefficient of filler dispersion, *λ*, and the coefficient of adhesion, *κ*, related to the strength of adhesion bond between filler and polymer matrix. Both parameters may vary between 0 and 1. When *κ* = 0 there is no adhesion between filler and matrix, but when *κ* = 1 the adhesion is perfect. Similarly, when *λ* = 0 there are the worst dispersion conditions, whereas with *λ* = 1 the dispersion level is optimal.

Three main cases can be identified based on the particular combination of these parameters (as shown in Figure 1): (1) When *λ* = *κ*, the variation of the modulus with filler-volume fraction was linear; (2) when *κ* < *λ* (for adhesion coefficient lower than the dispersion coefficient), the variation of the modulus was nonlinear and followed an upward concave curve; and (3) finally, for *κ* > *λ* (for adhesion coefficient higher than the dispersion coefficient), the variation of the modulus with filler volume fraction followed a convex curve. In this case, there was a strong effect of agglomeration, leading to a decrease in modulus with *C_f_*.

Then, two new parameters, *L* and *K*, can be introduced: $L=\frac{\lambda }{\lambda +\kappa }$ and $K=\frac{\kappa }{\lambda +\kappa }$. Obviously, *L* is proportional to the dispersion coefficient, *λ*. Thus, it is related to the percentage contribution of the filler dispersion within the polymer matrix to the overall property of the composite evaluated at the time, and is called *degree of dispersion*. Accordingly, *K* is proportional to the filler-matrix adhesion coefficient *κ*. Thus, it is related to the percentage contribution of the adhesion between filler and matrix to the overall property of the composite evaluated at the time, and is called *degree of adhesion*.

Evidently, the sum of these two parameters (*L* + *K*) is equal to 1. Also, if *δ*_1_ = *P*_1_ − *P_m_* and *δ*_2_ = *P*_2_ − *P_m_*, then
(11)$$L=\frac{\delta_{2}}{\delta_{2}+\delta_{1}\left(C_{2}/C_{1}\right)}$$
and
(12)$$K=\frac{\delta_{1}}{\delta_{1}+\delta_{2}\left(C_{1}/C_{2}\right)}$$
and *K/L* = (*δ*_1_/*δ*_2_)/(*C*_1_/*C*_2_).

At this point, it is important to note that both the degree of dispersion *L* and the degree of adhesion *K* depend upon the composite property computed at a specific time and they are varied both in the range from 0 to 1, so that *K* + *L* = 1. For the case of hydrothermal aging *E_c_* = *f*(*t*) and it is assumed that the property variation with the duration of environmental aging can be described as a quadratic polynomial:(13)$$E_{t}=At^{2}+Bt+E_{0}$$
where *E* and *E*_0_ represent the elastic moduli of each material studied during and before the environmental aging, accordingly. Thus, for two experimental points, (*t*_1_, *E*_1_) and (*t*_2_, *E*_2_), the following expressions could be written:(14)$$E_{1}=At_{1}{}^{2}+Bt_{1}+E_{0}$$
(15)$$E_{2}=At_{2}{}^{2}+Bt_{2}+E_{0}$$

Thus, by following the same procedure as described in [15], the two coefficients characterizing the degree of adhesion *K* and degree of dispersion *L* can be rewritten as:(16)$$K=\frac{E_{1}-E_{0}}{\left(E_{1}-E_{0})+(E_{2}-E_{0})\cdot (t_{1}/t_{2}\right)}$$
(17)$$L=\frac{E_{2}-E_{0}}{\left(E_{2}-E_{0})+(E_{1}-E_{0})\cdot (t_{2}/t_{1}\right)}$$

## 3. Results and Discussion

### 3.1. Microstructural Characterization

The microstructural characterization was performed to identify the appearance of agglomerates in the epoxy resin filled with CNT, CNF, and HN. The raw data were the areas of individual microsized particles (0.1–10 μm^2^) obtained from three independent optical micrographs. Due to limitations of optical microscopy, smaller particles were not registered and, therefore, were further disregarded by the analysis. The same filler content (0.1 wt. %) and magnification (×50) of the microscope were kept constant for all materials to assure the constant characteristic area for the analysis. The representative optical micrographs for the investigated materials and distribution by area are shown in Figure 2a,b, respectively.

Obviously, all materials had some amount of agglomerates since, according to Figure 2b, there was bimodal distribution of the area with characteristic peaks at approx. 0.2 and 2.0 µm^2^. By assuming that filler particles and their associated agglomerates had spherical shape, the radius of such individual particles was evaluated to be about 250 and 790 nm, accordingly (see Figure 2b). The appearance of two peaks for size (area) distribution indicates that there were two characteristic sizes of the filler particles for all systems (with single and hybrid nanofillers): Separated filler particles of radius approx. 250 nm and their associated agglomerates of size approx. 790 nm. By evaluating the average area of individual CNT 9.5 nm × 1.5 µm = 0.01 µm^2^ and CNF 130 nm × 20 µm = 2.6 µm^2^ using data from Table 1, it can be supposed that the particles for both single and hybrid nanofillers were dispersed rather effectively. According to Figure 2b, the number of filler particles registered for HN was the largest of all materials in the lowest area of 0.10 µm^2^. Consequently, it can be concluded that mixing conditions were the same efficiency for all materials studied.

### 3.2. Characterization before Hydrothermal Aging

#### 3.2.1. Flexural Properties

The three-point bending tests were carried out for all investigated materials before the hydrothermal aging to get a reference point for the analysis of hydrothermal aging effects on the flexural characteristics of the materials studied. The flexural stress in outer fibers at midpoint was evaluated by using the following equation
(18)$$\sigma =\frac{3PL}{2ah^{2}}$$

Here, *P* is the force applied at a given point for the force-deflection curve, *L* is the support span, and *a* and *h* are the width and the thickness of the specimen, respectively. The flexural strain in the outer surface at midpoint was evaluated by using the formula
(19)$$\varepsilon =\frac{6dh}{L^{2}}$$
where *d* is the maximum midpoint deflection of the specimen’s central section.

The flexural strength was considered as the maximal stress in the stress-strain curve, but the elastic modulus was evaluated by using stress-strain curves and evaluating the slope of a secant line for the strains 0.05% and 0.25%, accordingly.

The characteristic stress-strain curves are given in Figure 3a for the epoxy resin and all PNC, while the flexural strength and elastic modulus of the materials with polynomial approximations are provided in Figure 3b. According to Figure 3, before hydrothermal aging, the flexural properties were very similar for the epoxy and the PNC. According to Figure 3b, a minimal increase in the elastic modulus by approx. 2% and a decrease of flexural strength by approx. 10% was obtained for the filled epoxy at 0.3 wt. % of CNF. Consequently, the considerable improvement of the flexural properties of epoxy due to addition of single and hybrid carbon nanofillers was not observed despite possible improvement of interfacial strength between the nanofillers and the epoxy, as well as increased value of the shear component of the elastic modulus [20]. Evidently, such minimal effects on the flexural properties could be attributed to relatively low content of carbon nanofillers ranging from 0.05 wt. % (CNT) up to 0.3 wt. % (CNF) in the epoxy resin and also some agglomeration of filler particles observed by optical microscopy (see Figure 1) leading to weak nanofiller-matrix interfacial interactions.

#### 3.2.2. Thermomechanical Properties

As reported in [5], DMA provides important information to identify changes in the chain mobility restriction of the polymer network, which can be analyzed both before and after hydrothermal aging. Figure 4 summarizes the data for all materials studied for the storage modulus (*E*’) and loss factor (tan*δ*) in relation to temperature.

According to Figure 4a, before the hydrothermal aging, the materials studied were characterized by similar small decrease of the storage modulus in the glassy region, a sudden reduction in the glass-rubbery transition zone, and almost the same value in the rubbery region [5]. Analogously, the tan*δ* in relation to temperature was low in the glassy and rubbery states and passed through a maximum in the glass-rubbery transition zone, denoting glass transition temperature (*T*_g_) shown in Figure 4b vs. filler content.

As shown in Figure 4b, before the hygrothermal aging, the glass transition temperature of the materials studied was slightly increased for higher contents of carbon nanofiller(s) similarly as for the elastic modulus (see Figure 3b). Such slight increase by few degrees (or almost the same result) can be explained by the existence of such oppositely acting factors as formation of interphase/interface on the nanoparticles’ surfaces and agglomerates, as well as difference in crystallinity and crosslink density of the epoxy resin and the PNC [6,11].

### 3.3. Hydrothermal Aging

For the test samples which were immersed in water, the kinetics of water absorption were described by using classical Fick’s model for two-dimensional moisture diffusion
(20)$$w(t)=w_{\infty }-4\frac{\left(w_{\infty }-w_{0}\right)}{\pi^{4}}\sum_{k,i=1}^{\infty }\frac{{\left(1-{\left(-1\right)}^{k}\right)}^{2}}{k^{2}}\cdot \frac{{\left(1-{\left(-1\right)}^{i}\right)}^{2}}{i^{2}}\exp\left(-\left({\left(\frac{\pi k}{h}\right)}^{2}+{\left(\frac{\pi i}{a}\right)}^{2}\right)Dt\right)$$

Here *w*(*t*), *w_0_*, and *w*_∞_ are time-varying, initial, and the equilibrium moisture contents of a specimen, *D* is the diffusion coefficient of the material, and *a* and *h* are the thickness and width of a specimen, accordingly. Two-dimensional modes of Fick’s model were preferred since the evaluation results by using 1D mode showed relevant deviation from the experimental ones. According to Figure 5a, the application of Fick’s model for 2D case allowed good description of experimental data and there was no need to apply more complicated models of anomalous moisture diffusion.

The fictitious diffusion coefficient was evaluated from the initial slope of the moisture absorption curve as a function of $\sqrt{t}$ by using the equation for one-dimensional case of moisture diffusion [21]
(21)$$D=\frac{\pi h^{2}}{16t}{\left(\frac{w(t)-w_{0}}{w_{\infty }-w_{0}}\right)}^{2}$$

The maximal value of moisture content obtained during the water absorption was considered as the equilibrium moisture content in the specimen. The total duration of water absorption tests was 4 weeks. During this time all material specimens reached equilibrium water content, which is shown in Figure 4b together with the diffusion coefficient in relation to filler content.

As shown in Figure 4b, the maximum values for the equilibrium moisture content and diffusion coefficient were obtained for the neat epoxy resin. By adding a small amount (0.05, 0.1, and 0.3 wt. %) of stiff and moisture impenetrable CNT, CNF, and HN particles, the equilibrium moisture content was slightly reduced (0.06%, 0.23%, and 0.01%). Furthermore, the diffusion coefficient of the PNC was maximally reduced by approx. 2% in comparison with corresponding value for the neat epoxy resin (0.011 mm^2^/h). This latter result can be attributed probably to the excellent quality, mechanical properties, and high moisture resistance of the epoxies used for aeronautical applications in liquid resin transfer moulding processes [6,19].

### 3.4. Characterization after Hydrothermal Aging

#### 3.4.1. Flexural Properties

Mostly due to complicated and synergistic effects of moisture and temperature, the internal damages can appear in polymers and polymer-based composites (e.g., delaminations, microcracks, volume defects, and pores) leading to the decrease of the service life of structural components made of polymers and composites [1,4,9,12]. The changes in the flexural properties, elastic modulus (Figure 6a), and strength (Figure 6b), due to hydrothermal aging caused by water absorption at 70 °C or solely heating at 70 °C, are shown graphically for all materials tested, revealing differently induced effects.

As shown in Figure 6, the elastic modulus of the epoxy and all PNC increased by approx. 7% after water absorption. This can be explained by post-curing of the epoxy resin as previously reported in [6]. Evidently, water absorption and heating (*3*) caused higher effect on flexural characteristics in comparison with solely heating (*2*). After water sorption, the most relevant effects for the flexural strength were found for the epoxy (−9.5%) and filled epoxy at 0.3 wt. % of CNF (−11.1%), while it was the smallest for the case of filled epoxy at 0.1 wt. % of HN (−6.1% after water absorption and +0.5% after heating). Moreover, after water absorption and heating, the maximal deformation of all materials dropped twice, implying that the post-curing had a more critical effect than plasticization [5,19,22].

Experimental data and PPM model predictions of the elastic modulus values in the case of aged specimens without water at 70 °C are provided in Table 2, while corresponding curves of modulus variation with aging time at 70 °C as well as curves showing the variation of the degrees of adhesion, *K*, and dispersion, *L*, for analogous materials as function of the filler concentration are shown in Figure 7a,b, respectively.

As observed, model predictions are in good relation to the appropriate experimental results showing a maximum deviation of less than 5%. As for the *K* and *L* variation (see Figure 7b) with filler concentration, they both remained almost constant and independent by the concentration value. Evidently, all PNC showed high filler-matrix adhesion, while the dispersion was extremely low for all filler concentrations. This behavior can be attributed to the formation of filler aggregates even from the early stages of reinforcement as also confirmed by bimodal distribution of the filler particle area, according to Figure 2b.

Adhesion and dispersion are two antagonistic phenomena. The higher the adhesion, the lower the dispersion is. Even for the lowest amount of inclusions incorporated in the matrix, the filler-matrix adhesion was high enough, keeping the degree of dispersion at very low levels. Therefore, it can be concluded that as the specimens were heated the resulting effect of heating was not strong enough to produce any change to either the adhesion or dispersion degrees.

In the following, PPM was applied to describe the variation of the elastic modulus of the investigated materials, which were subjected to water absorption at 70 °C. Generally, the immersion of composite materials in water leads to the degradation of their thermomechanical properties [23,24], which is related to plasticization induced by moisture, as well as micromechanical damage and matrix cracking, thus, forming extra cavities to be filled with water.

Temperature effect of the humid media is also relevant issue when studying the variation of the mechanical properties of the polymers and polymer composites during water absorption when free volume and the molecular mobility of polymer macromolecules increase. Such enhancement of molecular mobility results in a breakdown of the hydrogen bonds between the water molecules and the active sites in the polymer matrix. Furthermore, the state of interfacial bond between polymer matrix and the filler is also crucial. Thus, perfect adhesion, which is related to complete continuity of stresses and displacements at the interface, is generally assumed in micromechanical models [23,24]. Though, in reality, this condition is hardly realized in the composites. Therefore, microcracks and voids to high extent affect the stress and strain fields appearing in the composites. A particular case of imperfect bonding is the agglomeration, which is dependent both on the state of filler dispersion and/or size of filler particles, in turn, controlled by the manufacturing process.

According to the above discussion, it follows that it is a complicated task to find the interrelation between sorption and mechanical data due to a large number of involved parameters. In addition, all above-mentioned mechanisms tremendously affected the degrees of filler-matrix adhesion and filler dispersion. In Table 3, both experimental values and respective model predictions for the elastic modulus are presented. For all materials, a good correlation between experimental and predicted values was observed. The same elastic modulus variation with time of immersion is plotted and the curves are presented in Figure 8a. This variation is linear, showing a very small variation with time of immersion (see also exact values in Table 3).

However, as shown in Figure 8b, in the case of water absorption there was a strong effect on both the degree of adhesion between filler and matrix, *K*, as well as on the degree of filler dispersion in the polymer matrix, *L*. This is owing to the combined effects of filler fraction, water absorption, and temperature. More precisely, for the specimens aged in water at 70 °C, at low filler concentrations, *K* and *L* degrees showed moderate values. As the filler content increased, an increase in filler-matrix contact area was developed, leading to a higher degree of adhesion and, respectively, to a lower degree of dispersion. At a certain filler content aggregates were created, thus decreasing the filler-matrix contact area while increasing the free volume (in the form of voids) creation, which was immediately filled with water molecules. Water then can become bound to network sites, causing swelling, which results in an abrupt disruption of matrix-filler interfacial bonds at the filler-matrix interphase leading to a subsequent decrease in filler-matrix degree of adhesion and a respective increase in the degree of dispersion.

Moreover, according to Figure 7a and Figure 8a, it was clear that, although the nanofiller relative content increased from 0.05 to 0.3 wt. %, i.e., by a factor of 6, the corresponding elastic modulus decreased. This could be attributed to the enhanced agglomeration at increasing filler content (see Figure 1) and, at the same time, to the significant amount of water molecules and voids which could concentrate at the filler-matrix interface. Both phenomena led to weak filler-matrix adhesion (*K* = 0). Due to the agglomeration effect, there was less filler matrix contact area available, while, due to water and voids interface concentration, there was no direct contact between filler and matrix. Also, the agglomerates formed at high nanofiller contents can be considered as micro-inclusions (not nano-) which, however, were dispersed into the bulk matrix individually (*L* = 1). Analogous results were provided for the elastic modulus and strength of the epoxy filled with nano- and microparticles of TiO_2_ in [15]. In addition to that, when adding nanofillers into a polymeric matrix, due to their nano-dimensions, they could “enter” into the polymers’ free volume preventing the polymerization procedure. This is equivalent to a “plastification” effect. Obviously, the higher the amount of nanotubes, the stronger is the “plastification” effect.

It should be noted that parameters *K* and *L* were different for the samples which were subjected to water absorption and heating (Figure 8b) and solely heating (Figure 7b). The greatest difference was observed for the filled epoxy at 0.3 wt. % of CNF. In the case upon thermal heating, the degree of adhesion *K* was equal to 0.88 but the dispersion parameter *L* was equal to 0.12. The reason for it can be associated to different induced effects of moisture and temperature on the structure and properties of the matrix/filler interphase. Obviously, owing to high surface area of the nanoparticles at high filler contents and, subsequently, high contact area between filler and matrix, almost all matrix material can be transferred into “modified matrix”, i.e., interfacial material [15]. Thus, for the filled epoxy at 0.3 wt. % of CNF potentially having very high content of interphase material such deviation may exist for the degree of adhesion and dispersion revealing that adhesion is more influenced by the action of temperature while the dispersion is more influenced by the action of moisture, while for the filled epoxy at 0.1 wt. % of HN the results were exactly the same after aging in water and without water (*K* = 0.84 and *L* = 0.16) indicating, however, that although the degree of adhesion was good, the degree of dispersion was relatively low. Hypothetically it can be summarized that the addition of hybrid nanofiller allowed minimizing “plastification” effect of the neat matrix due to addition of nanoparticles with similar aspect ratio but characterized by a surface area of 10 times different (see Table 1).

#### 3.4.2. Thermomechanical Properties

The effect of hydrothermal aging in water and without water on the thermomechanical properties is shown in Figure 9 for all investigated materials. Also, in this case as above, before the hydrothermal aging (see Figure 4) the representative curves for storage modulus as a function of temperature were similar for the epoxy and the PNC. The position of tan*δ* curves of the investigated materials indicated that *T*_g_ for the epoxy resin was higher by ~3 degrees compared to the PNC upon hydrothermal aging either with or without water. Also, it may be noticed from Figure 4 and Figure 9 that the height of tan*δ* curves for the epoxy resin decreased upon hydrothermal aging, indicating a microstructural reorganization and a subsequent decrease of polymer chain mobility and free volume. Similarly, almost no change in the height of tan*δ* curves due to cyclic moisture absorption, could be appreciated for the epoxy filled with CNT [6]. This was explained, mainly, by the reduction in free volume and enhanced physical aging owing to lower segmental mobility of the polymer chains and different cure kinetics of the polymers and PNC prior to environmental aging [25,26].

From Figure 9, it can be noticed that after hydrothermal aging the spectra for tan*δ* curves are wider for the epoxy in comparison with the epoxy filled with CNT, CNF, and NH. Such broadening and sometimes even splitting into two separate peaks upon exposure to water at different temperatures is widely discussed in the literature [5,11,27]. Mostly, it is attributed to differential plasticization conditions in the glass-rubbery relaxation, which is closely related to the nonhomogeneity of the system. Thus, the existence of both microstructural plasticized and less- (or even non-) plasticized regions may result in high distribution of polymer chains by the length and, therefore, leading to broadening of tan*δ* curves for the moistened polymers.

It can be also noted that the growth of glass transition temperature of the epoxy upon hydrothermal aging associated to post-curing and decrease of the free volume correlated well with the increase of elastic modulus (see Figure 7a and Figure 8a). For the case of the epoxy filled with CNT, CNF, or HN, there was minimal effect on the glass transition temperature before/after hydrothermal aging and this result agrees well with the literature data being attributed to several opposite factors, such as formation of interphase/interface and agglomeration of the nanoparticles, as well as differences in the crystallinity and the crosslink density of the epoxy resin and the PNC [6,11,28].

## 4. Conclusions

The effect of hydrothermal aging on the flexural and thermomechanical characteristics of the epoxy and epoxy filled with different carbon nanofillers was evaluated in the current study. Apparently, before the hydrothermal aging the addition of carbon nanofillers did not result in the improvement of the flexural and thermomechanical properties, which was explained by comparably low filler fraction in the epoxy resin and also agglomeration of nanofillers’ particles, causing weak interfacial interactions between the epoxy matrix and carbon nanofillers.

Water absorption at 70 °C and thermal heating at 70 °C differently affected the investigated materials, and the environmental aging in water had more influence on flexural strength and elastic modulus if compared to solely heating. The correlation between the growth in the glass transition temperature and the elastic modulus of the epoxy was found after hydrothermal aging, indicating on possible post-cure. For the case of all PNC, minimal effect on the glass transition temperature before and after hydrothermal aging was observed, indicating improved stability to environmental factors if compared to the epoxy resin.

The property prediction model was applied for the description of the elastic modulus of all materials due to hygrothermal aging during water absorption and heating, and solely heating. It provided very good prediction for the variation of the elastic modulus as a function of time for the epoxy and epoxy filled with all carbon nanofillers showing a maximum deviation less than 5%. All PNC showed high filler-matrix adhesion while the dispersion was relatively low. Also, interesting results were obtained for PPM parameters, degrees of adhesion and dispersion, revealing differently induced effects of moisture and temperature on the properties of the matrix/filler interphase. For the epoxy resin filled with 0.1 wt. % of HN, the results were the same after hydrothermal aging in water and without water (*K* = 0.84 and *L* = 0.16). However, revealing that, though the degree of adhesion was good, the degree of dispersion was relatively low.

Based on experimental and theoretical results, the epoxy filled with 0.1 wt. % of HN was characterized by the lowest effect of both temperature and moisture on material characteristics, along with the lowest sorption characteristics, indicating improved stability to the environmental factors, which is particularly critical for outdoor applications.

## Figures and Tables

**Figure 1 polymers-12-01153-f001:**
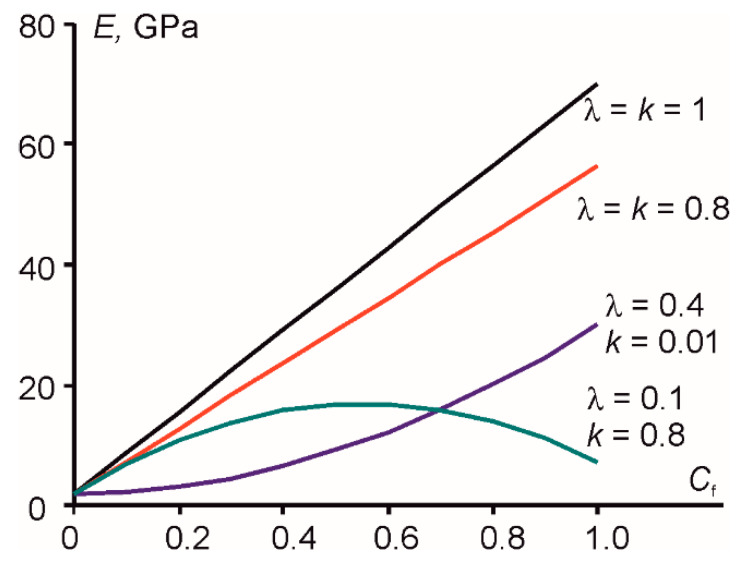
Elastic modulus of the polymer nanocomposites predicted by property prediction model and Equation (10) for different model parameters (indicated next to the curves, elastic modulus of the filler *E_f_* = 70 GPa, elastic modulus of the matrix *E_m_* = 2 GPa).

**Figure 2 polymers-12-01153-f002:**
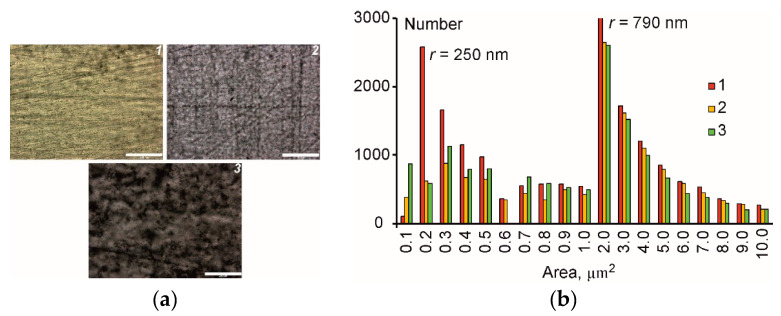
(**a**) Representative optical images (scale: 200 μm); (**b**) size distribution for the epoxy filled with CNT (*1*), CNF (*2*), and HN (*3*). The filler content was 0.1 wt. % for all PNC.

**Figure 3 polymers-12-01153-f003:**
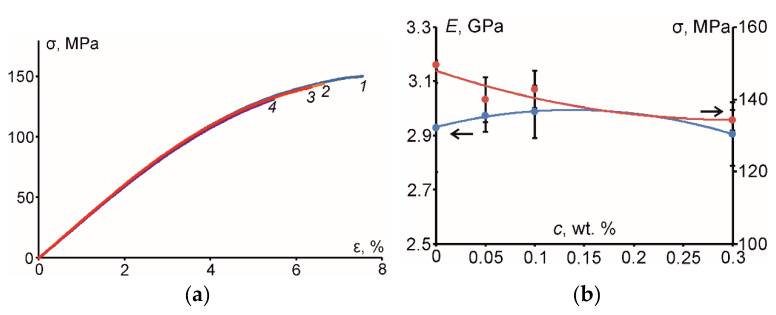
(**a**) Representative stress-strain curves for the epoxy (*1*) and epoxy filled with 0.05 wt. % of CNT (*2*), 0.1 wt. % of HN (*3*), and 0.3 wt. % of CNF (*4*); (**b**) elastic modulus and flexural strength of the epoxy and PNC vs. filler weight content (dots—experimental values, lines—polynomial approximations).

**Figure 4 polymers-12-01153-f004:**
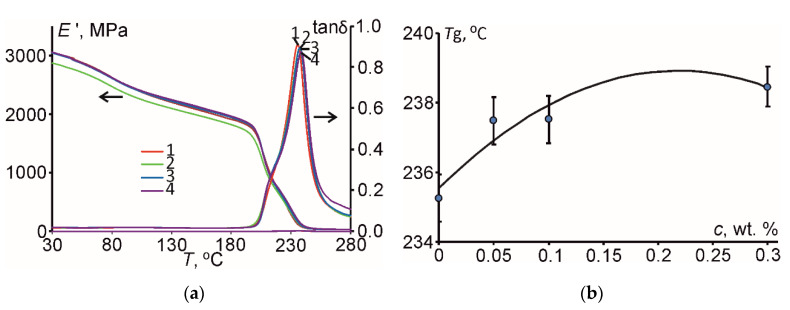
(**a**) Storage modulus (*E*’) and loss factor (tan*δ*) of the epoxy (*1*) and epoxy filled with 0.05 wt. % of CNT (*2*), 0.1 wt. % of HN (*3*), and 0.3 wt. % of CNF (*4*) vs. temperature before the hydrothermal aging; (**b**) glass transition temperature (*T*_g_) vs. filler weight fraction for the materials studied (dots—experimental data, line—polynomial approximation).

**Figure 5 polymers-12-01153-f005:**
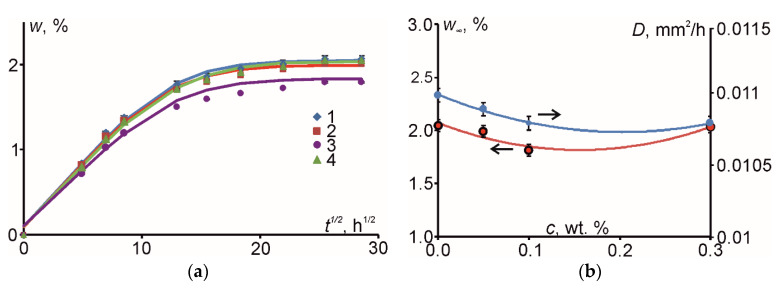
(**a**) Kinetics of water sorption of the epoxy (*1*) and epoxy filled with 0.05 wt. % of CNT (*2*), 0.1 wt. % of HN (*3*), and 0.3 wt. % of CNF (*4*) (dots-experimental data, lines-evaluation by Equation (20); (**b**) diffusion coefficient and equilibrium moisture content as function of nanofiller’s (-s’) weight content (dots-experimental data, lines-polynomial approximations).

**Figure 6 polymers-12-01153-f006:**
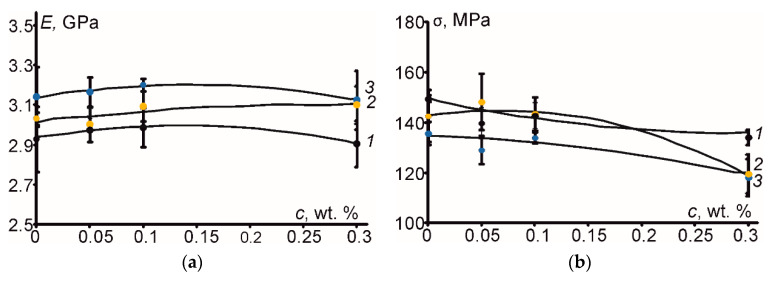
(**a**) Elastic modulus and (**b**) flexural strength before (*1*) and after 4 weeks of the hydrothermal aging vs. carbon nanofiller’s (-s’) weight content heated at 70 °C (*2*) or immersed in water at 70 °C (*3*) (dots—experimental data, lines—polynomic approximations).

**Figure 7 polymers-12-01153-f007:**
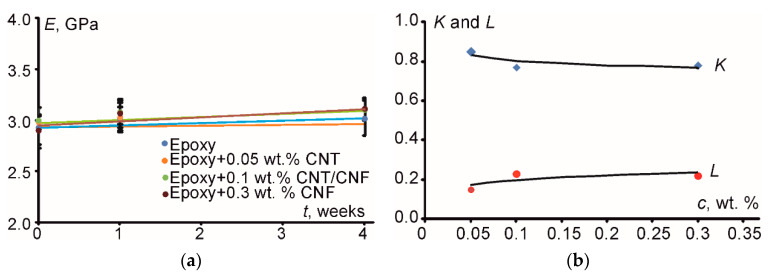
Experimental results and respective theoretical predictions by using PPM for (**a**) the elastic modulus variation of the investigated materials (indicated in the legend) with aging time at 70 °C and (**b**) variation of the degrees of adhesion (*K*) and dispersion (*L*) with filler concentration for specimens aged at 70 °C (dots—experimental data, lines—PPM prediction).

**Figure 8 polymers-12-01153-f008:**
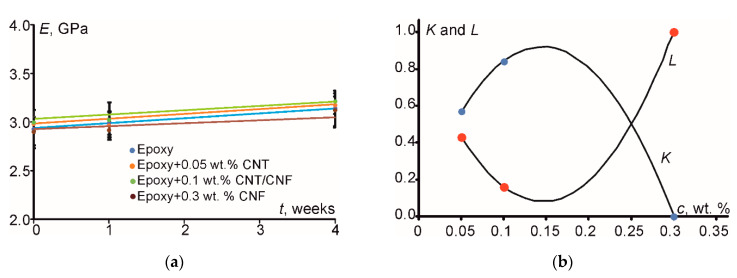
Experimental results and theoretical prediction by using PPM for (**a**) the elastic modulus variation of the investigated materials (indicated in the legend) with time of immersion in water at 70 °C and (**b**) variation of the degrees of adhesion (*K*) and dispersion (*L*) with filler concentration for specimens aged at 70 °C (dots—experimental data, lines—PPM prediction).

**Figure 9 polymers-12-01153-f009:**
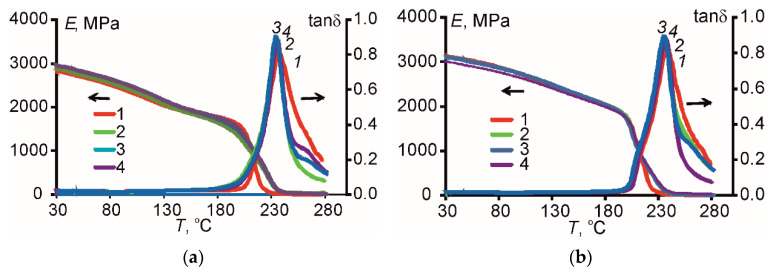
Storage modulus (*E*’) and loss factor (tan*δ*) of the epoxy (*1*) and epoxy filled with 0.05 wt. % of CNT (*2*), 0.1 wt. % of HN (*3*), and 0.3 wt. % of CNF (*4*) vs. temperature (**a**) after water absorption at 70 °C and (**b**) after heating at 70 °C.

**Table 1 polymers-12-01153-t001:** Characteristics of the carbon nanotubes (CNT) [16] and carbon nanofibres (CNF) [17] according to manufacturers’ datasheets.

Material Characteristics	CNT	CNF
Average diameter, nm	9.5	130
Average length, μm	1.5	20
Aspect ratio	158	153
Surface area, m^2^/g	250–300	24
Density, g/cm^3^	1.75	1.90

**Table 2 polymers-12-01153-t002:** Experimental and predicted elastic modulus for the specimens heated at 70 °C.

Time, Weeks	Epoxy		Epoxy + 0.05 wt. % CNT		Epoxy + 0.1 wt. % CNF		Epoxy + 0.3 wt. % CNT/CNF	
	Exp.	Pred.	Exp.	Pred.	Exp.	Pred.	Exp.	Pred.
0	2.93 ± 0.16	2.93	2.97 ± 0.06	2.93	2.99 ± 0.10	2.93	2.91 ± 0.12	2.91
1	3.02 ± 0.07	3.02	3.03 ± 0.07	3.03	3.07 ± 0.10	3.07	3.07 ± 0.11	3.07
4	3.03 ± 0.03	3.13	3.09 ± 0.07	3.18	3.09 ± 0.07	3.29	3.10 ± 0.09	3.31
			*K* = 0.89 *L* = 0.11		*K* = 0.84 *L* = 0.16		*K* = 0.88 *L* = 0.12	


**Table 3 polymers-12-01153-t003:** Experimental and predicted elastic modulus for the specimens stored in water at 70 °C.

Time, Weeks	Epoxy		Epoxy + 0.05 wt. % CNT		Epoxy + 0.1 wt. % CNF		Epoxy + 0.3 wt. % CNT/CNF	
	Exp.	Pred.	Exp.	Pred.	Exp.	Pred.	Exp.	Pred.
0	2.93 ± 0.16	2.93	2.97 ± 0.06	2.93	2.99 ± 0.12	2.93	2.91 ± 0.10	2.93
1	2.99 ± 0.10	2.94	2.97 ± 0.12	3.01	3.02 ± 0.08	3.22	2.93 ± 0.15	2.89
4	3.14 ± 0.15	3.14	3.17 ± 0.07	3.17	3.20 ± 0.15	3.18	3.13 ± 0.03	2.98
			*K* = 0.57 *L* = 0.43		*K* = 0.84 *L* = 0.16		*K* = 0.00 *L* = 1.00	

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
