# Peer review of "Hydrothermal Aging of an Epoxy Resin Filled with Carbon Nanofillers"

_polymers, 2020, doi:10.3390/polym12051153_

Round 1
Reviewer 1 Report
Summary
This paper presents a study on the effect of hydrothermal ageing on the flexural properties of an epoxy resin filled with carbon nanofillers. Three different nanofillers have been used to obtain the polymer nanocomposites (PNC). Two different regimes of ageing were applied: heating up to 70ºC, and heating at 70ºC with water absorption. Then, three-point bending tests and dynamic mechanical tests were conducted before and after applying the two ageing procedures. In parallel, theoretical predictions were performed by the property prediction model (PPM).
Mayor Concerns
Regarding to the three-point bending tests, the results (flexural modulus and flexural strength) are presented in Figure 6 (the exact values for the flexural modulus are collected in Tables 2 and 3). The main conclusion stated by the authors is that “as shown in Figure 6, the flexural modulus of the epoxy and all PNC increased by approx. 7% after water absorption. This can be explained by post-curing of the epoxy resin as previously reported in [6]”. It seems that this effect occurs independently on adding or not nanofillers. Respect the other material property, the flexural strength, Figure 6 b is not clear: the black points (results before ageing) are sometimes above the blue and yellow points (results after ageing), or between or below. So I am not sure about the conclusion obtained by the authors: “According to the results obtained it can be deduced, the addition of hybrid carbon nanofiller to the epoxy resin allows minimizing moisture and temperature effects over the flexural properties of the neat system”. In my opinion this study confirms the post-curing effect on the flexural modulus and it is not possible to obtain an evident conclusion about the behaviour of the flexural strength.
The thermomechanical properties studied through the dynamic mechanical tests are the storage modulus and the loss factor that indicates the glass transition temperature. The results are shown in figure 4 (before ageing), figure 9a (after water absorption at 70ºC) and figure 9b (after heating at 70ºC). Two conclusions are extracted. The first one: “The position of tanδ curves of the investigated materials indicated that Tg for the epoxy resin was higher by ~3 degrees compared to the PNC upon hydrothermal ageing either with or without water”. The second conclusion, as stated by the authors is the next one: “From Figure 9, it can either noticed that after hydrothermal ageing the spectra for tand curves are wider for the epoxy in comparison with the epoxy filled with CNT, CNF and NH. Such broadening and sometimes even splitting into two separate peaks upon exposure to water at different temperatures, is widely discussed in the literature [5, 11, 27]”
The prediction of the Property Prediction modelling (PPM) are according to the experimental results, but this model has been “previously successfully applied for the prediction of flexural modulus and strength of epoxy resin filled with micro- and nanoparticles of TiO2 as a function of filler concentration [15]”.
In summary, this is a paper well written, with a rigorous experimental procedure, but the main conclusions are yet described in other works. If the authors think that the work merits publication should highlight their contributions with respect other previously published works.
Minor Concerns
- I do not understand the reason of using the term “flexural properties”. I think that are “elastic properties” obtained from a “flexural” test.
- A brief explanation about the reasons of using the 0,05%, 0,1% and 0,3% wt for the three nanofillers should be given.
- Why the microstructural characterization is obtained with 0,1 % wt? This composition is different from those used in the paper
- I suggest to order the values in the tables according to the axis-x of the figures: epoxy, epoxy + 0,05% wt, epoxy + 0,1% wt and epoxy + 0,3% wt
- What is the meaning of Pf in PPM?
- The quality of the figures should be improved. There are difficult to understand.
Reviewer 2 Report
The manuscript deals with effect of environmental exposure on the durability of epoxy based nanocomposites including carbonaceous fillers. At this regard, three-point bending and dynamic-mechanical characterizations were performed during and after water absorption at 70°C and/or thermal heating up to 70 °C of epoxy resins filled with carbon nanotubes (CNT), carbon nanofibers (CNF) with almost the same aspect ratio or both (hybrids).
Moreover, authors estimated the influence of environmental effects on flexural modulus of materials investigated, as the filler content changes, with the help of a Property Prediction Model (PPM), already validated for epoxy systems including TiO2 particles.
Model predictions in terms of the variation of two parameters as degree of adhesion and dispersion as a function of the filler content resulted to be in good relation to the appropriate experimental results.
All results, dependent on the characteristics of the phases involved but also on a compromise between plasticization and post-cure phenomena triggered by the applied hydrothermal conditioning, are clearly described and interpreted and allow to acquire useful knowledge for the design of these materials especially if intended for outdoor applications.
Appreciating the novelty and scientific relevance of the research, conducted inter alia with appropriate procedures, the paper is recommended for publication.
Author Response
Thank you very much for the positive evaluation of our work.
Round 2
Reviewer 1 Report
I would thank the authors for the effort in improve the paper. I am satisfied with the reviions